# PASS: Pruning Attention Heads with Almost-sure Sparsity Targets

**Dujian Ding**  *dujian.ding@gmail.com*
*Department of Computer Science*
*the University of British Columbia*

**Ganesh Jawahar**  *ganeshjwhr@gmail.com*
*Google DeepMind*

**Laks V.S. Lakshmanan**  *laks@cs.ubc.ca*
*Department of Computer Science*
*the University of British Columbia*

**Reviewed on OpenReview:** *https://openreview.net/forum?id=S4duStTKGL*

## Abstract

Transformer models have been widely used to obtain high accuracy values in multiple fields including natural language processing (NLP), computer vision, and more. This superior performance typically comes at the expense of substantial computational overhead. Multi-head attention is the key factor in the success of Transformer models that has been found to be computationally expensive. Significant research effort has been devoted to improving attention compute efficiency by pruning redundant attention heads. A widely adopted paradigm is to jointly learn a set of gate variables and apply thresholds on gate values to prune heads. Previous work shows a high level of sensitivity to threshold tuning which can limit subnetwork performance and prevent them from wider adoption in practice. We propose the notion of *almost-sure* sparsity to overcome this limitation and develop a generic framework for **P**runing with **A**lmost-**S**ure **S**parsity (PASS) targets over attention heads. To further boost efficiency, we design a novel technique, *concentrator*, based on which we develop PASSCONC (**PASS** with **CONC**entrator). We also present a simple-yet-effective strategy to further improve subnetwork performance by clipping and selectively reopening learned gates. We investigate PASS and PASSCONC on two widely studied architectures: encoder-decoder (ED) Transformer and encoder-only Transformer (e.g., BERT-base). Experiments on IWSLT14 German-to-English translation and GLUE benchmark tasks demonstrate that our approaches outperform the SOTA by achieving up to 1.33 higher BLEU scores, 1.44% higher accuracy, and 60% higher attention speedups.

## 1 Introduction

Transformer models (Vaswani et al., 2017) have become a lead force in the study of natural language processing (NLP), computer vision, information retrieval, and other domains (Hua et al., 2024; 2023; Asai et al., 2024; Darcet et al., 2024; Ding et al., 2024). As Transformers grow deeper and larger, however, their application on longer contexts remains challenging because attention computation, which is at the heart of Transformer architectures, is of quadratic time and memory complexity with respect to the input length (Dao et al., 2022). For example, Wang et al. (2020a) observed that attention computation typically accounts for over 50% end-to-end latency of a GPT-2 model on multiple hardware platforms.

Significant research efforts have been devoted to improving attention computation efficiency from two orthogonal perspectives: *reducing attention complexity* and *pruning attention heads*. As a successful attempt

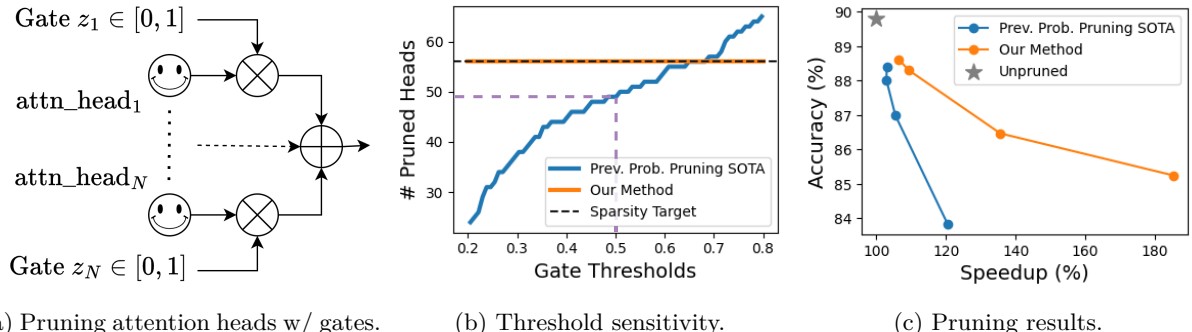

(a) Pruning attention heads w/ gates.   (b) Threshold sensitivity.   (c) Pruning results.

Figure 1: (a) Pruning attention heads by learning a set of gate variables. Gates take values from $[0, 1]$ and apply to attention heads before summation. Heads of low gate values are more likely to be pruned. (b) With a Transformer model of 72 heads and a sparsity target aiming to prune 56 heads, previous probabilistic pruning SOTA (Xia et al., 2022) is highly sensitive to gate threshold tuning (a 0.5 threshold only prunes 49 heads) while our approach consistently achieves the desired sparsity target (e.g., pruning 56 heads) in a threshold-independent manner. (c) On GLUE benchmark tasks (Wang et al., 2018), our approach achieves up to 1.4% higher average accuracy and 60% higher speedups than previous probabilistic pruning SOTA. Points in the plot represent the accuracy-speedup trade-offs achieved by different approaches under various sparsity targets. Unpruned model performance is also included for reference purpose.

at *reducing attention complexity*, sparse attention (Roy et al., 2021; Tay et al., 2020; Child et al., 2019) focuses on sparsifying the attention distribution over tokens for each head to improve efficiency. Linformer (Wang et al., 2020b) reduces the attention compute complexity from $O(N^2)$ to $O(N)$ with low-rank matrix approximation. FlashAttention (Dao et al., 2022; Dao, 2023) focuses on memory access efficiency in attention computation and achieves linear memory complexity by exploiting the asymmetric GPU memory hierarchy and minimizing unnecessary data transfers. However, implementing these approaches introduces extra challenges, particularly in rewriting the attention arithmetic operations as well as the underlying CUDA kernels to improve hardware utilization, which prevents them from wider adoption in practice.

The second line of work focuses on *pruning attention heads* (Voita et al., 2019; Li et al., 2021; Xia et al., 2023) to achieve significant inference speedups without changing the arithmetic operations of attention modules and therefore can be applied to a majority of Transformer models with only minor training/fine-tuning configuration changes. These works utilize the fact that properly trained Transformers are highly over-parameterized, and study how to extract efficient subnetworks by removing redundant heads without significant performance drops. A widely adopted paradigm is to jointly learn a set of trainable gate variables for each attention head, as shown in Figure 1a. At test time, attention heads associated with low gate values are pruned subject to predefined thresholds or sparsity targets. Typically, Li et al. (2021) achieves user-specified attention head sparsity by iteratively applying the Gumbel-softmax trick (Gumbel, 1954) to select the top-$K$ most important attention heads for given sparsity targets. At each training iteration, however, only selected attention heads get updated by the training optimizer (e.g., Adam (Kingma & Ba, 2015)) which prevents models from being trained with more heads in early training stages and limits the final subnetwork performance. Voita et al. (2019) and Xia et al. (2022) overcome this limitation by allowing all heads to participate in the training process and achieve sparsity in a probabilistic manner. Specifically, Voita et al. (2019) learns the probability distribution for gate values and sparsifies the models by regularizing the gate closing probability (the likelihood that gate variables equal 0). Xia et al. (2022) follows a similar probabilistic forumulation and achieves the *target sparsity in expectation* by using the *lagrangian multiplier* (Wang et al., 2020c) to explicitly enforce sparsity constraints on attention heads. At test time, both Voita et al. (2019) and Xia et al. (2022) use a threshold (e.g., 0.5) on the gate closing probabilities to determine if an attention head can be pruned confidently. In practice, however, these probabilistic pruning approaches suffer from two limitations. Firstly, setting correct thresholds is challenging and a mistakenly chosen threshold can lead to failed sparsity targets, as illustrated in Figure 1b. Secondly, thresholding gate variables at test time may bring significant architecture changes to Transformer models and lead to decayed subnetwork performance.

On GLUE benchmark tasks (Wang et al., 2018), we observe that previous probabilistic pruning SOTA (Xia et al., 2022) may lead to drastic performance decay even with correctly calibrated gate thresholds, while our approach achieves up to 1.4% higher average accuracy and 60% higher speedups (see Figure 1c).

Motivated by these limitations, in this work we propose a novel pruning approach, **P**runing with **A**lmost-**S**ure **S**parsity targets (PASS), to achieve good subnetwork performance by involving all heads during model training and prune models to the desired sparsity levels in a threshold-independent manner. Similar to Voita et al. (2019) and Xia et al. (2022), PASS takes the probabilistic pruning formulation but explicitly enforces all gate variables to diverge as the pruning process proceeds. The most important observation is that, if the closing probabilities for all gate variables approach either 0 or 1 with respect to the sparsity targets after training, we can easily extract the desired sparse subnetworks by introducing no further changes in attention compute at test time and therefore retain the subnetwork performance. We use the well-established notion "almost surely" in probability theory (Jacod & Protter, 2003) and propose the notion of *almost-sure* sparsity (see Section 3.1). We say a gate can be *closed almost surely* if the corresponding gate closing probability equals 1. In PASS, we express the sparsity targets in terms of the almost sure sparsity and extract the desired subnetworks by enforcing the sparsity constraints through regularization techniques in model training process (see Section 3.2). To push the envelope on inference efficiency, we propose a novel technique, *concentrator*, based on which we develop PASSCONC (**PASS** with **CONC**entrator), as discussed in Section 3.3. Observing the gradient vanishing problem in the gate variable training process, we present a simple-yet-effective strategy to clip and selectively reopen learned gates which leads to improved subnetwork performance (see Section 3.4). We evaluate our methods with encoder-decoder (ED) Transformer models and BERT models on IWSLT14 German-to-English translation (Cettolo et al., 2014) and GLUE benchmark tasks (Wang et al., 2018). We explore the Pareto front between model performance and inference efficiency for subnetworks identified by PASS, PASSCONC, and recent work (Li et al., 2021; Xia et al., 2022; Voita et al., 2019). Experiments show that PASS and PASSCONC outperform all baselines across a majority of experiment settings, by identifying subnetworks of higher speedups and better model performance (see Section 4). For example, on GLUE benchmark tasks, PASSCONC achieves a 185.2% attention speedup on average, which is 60% higher than all baselines, while providing even higher accuracy. This observation suggests that PASS and PASSCONC are capable of identifying subnetworks with high model capability and can be applied to resource-limited applications to achieve good performance-efficiency trade-offs.

In this work, we make the following contributions.

1. We propose a novel notion of *almost-sure* sparsity and develop an effective model pruning framework PASS to prune models to specified *almost-sure* sparsity levels on attention heads.

2. We propose a novel technique, *concentrator*, to further push the envelope on model inference efficiency and develop PASSCONC.

3. We present a simple-yet-effective strategy to further improve subnetwork performance by clipping and selectively reopening learned gates.

4. We evaluate PASS and PASSCONC on ED Transformer and BERT models with well established NLP tasks. Experiments show that PASS and PASSCONC outperform baselines by obtaining significant efficiency improvements and better performance-efficiency trade-offs.

## 2 Preliminaries

A frequently encountered task in machine learning is to find the model that minimizes the negative log-likelihood of an observed dataset, which can be formulated as follows,

$$\theta^* = \arg\min_{\theta} -\log P(D|\theta) \tag{1}$$

where $D$ is an observed dataset and $\theta = \{\theta_1, \theta_2, \cdots, \theta_{|\theta|}\}$ stands for the parameters of a parameterized model (e.g., a neural network). In real-world applications, we typically have model sparsity constraints to prevent high inference latency or reduce memory footprints (Gupta & Agrawal, 2022). A recent line of work (Louizos

et al., 2017; Voita et al., 2019) pursues this goal by training gate variables, $z = \{z_1, z_2, \cdots, z_{|\theta|}\}$, jointly with parameters, $\theta$. Each $z_i \in \mathbf{z}$ has support $[0,1]$. The objective function Eq. 1 can be re-parameterized as,

$$\theta^* = \arg\min_{\theta} -\log P(D|\theta \odot \mathbf{z}) \tag{2}$$

where $\odot$ indicates component-wise multiplication between network parameters $\theta$ and the gate variables $\mathbf{z}$. Typically, $\mathbf{z}$ is a latent variable following the posterior distribution $p(\mathbf{z}|D)$, which reflects the user-defined sparsity constraints. The probabilistic pruning approaches (Voita et al., 2019; Xia et al., 2022) aim to optimize the expected likelihood over the posterior distribution of the gate variables $\mathbf{z}$,

$$\theta^* = \arg\min_{\theta} -\log \mathbb{E}_{p(\mathbf{z}|D)}[P(D|\theta \odot \mathbf{z})] \tag{3}$$

The objective function described by Eq. 3 is mathematically intractable when the posterior $p(\mathbf{z}|D)$ is *a priori* unknown. As an attempt to tackle such intractability, we can first derive the *evidence lower bound* of the log-likelihood in Eq. 3 which is a widely used technique in previous variational inference work (Vahdat et al., 2018a;b). Since we are interested in minimizing the negative log-likelihood, it gives us an upper bound for the objective in Eq. 3 [1],

$$-\log \mathbb{E}_{p(\mathbf{z}|D)}[P(D|\theta \odot \mathbf{z})] \leq -\mathbb{E}_{q(\mathbf{z};\Phi)}[\log P(D|\theta \odot \mathbf{z})] + KL\left(q(\mathbf{z};\Phi)||p(\mathbf{z}|D)\right) \tag{4}$$

where $q(\mathbf{z};\Phi)$ is an *approximate posterior* distribution parameterized by $\Phi = \{\phi_1, \phi_2, \cdots, \phi_{|\theta|}\}$. Detailed derivation can be found in Appendix A.1. Minimizing this upper bound with respect to $q(\mathbf{z};\Phi)$ results in $q(\mathbf{z};\Phi) = p(\mathbf{z}|D)$ and turns the inequality into an equality (Beal, 2003). By denoting this upper bound as $\mathcal{L}(\theta, \Phi)$, we can then formulate the learning problem as,

$$\mathcal{L}(\theta, \Phi) = -\mathbb{E}_{q(\mathbf{z};\Phi)}[\log P(D|\theta \odot \mathbf{z})] + KL(q(\mathbf{z};\Phi)||p(\mathbf{z}|D))$$
$$\theta^*, \Phi^* = \arg\min_{\theta, \Phi} \mathcal{L}(\theta, \Phi) \tag{5}$$

We aim to jointly learn the optimal network parameters $\theta^*$ and the distribution of gate variables, $\Phi^*$, by minimizing the upper bound $\mathcal{L}(\theta, \Phi)$.

The foregoing analysis gives a generic framework to enforce sparsity over neural models which is agnostic to the underlying network structures. To prune attention heads, all we need is to assign each head a gate variable and solve Eq. 5 with $\mathbf{z} = \{z_1, z_2, \cdots, z_{|\mathcal{H}|}\}$, where $\mathcal{H}$ is set of all attention heads (see Figure 1a).

## 3 Methodology

### 3.1 Almost-sure Sparsity

The KL-divergence term in Eq. 5 is mathematically intractable when the true posterior $p(\mathbf{z}|D)$ is unknown. A line of work (Voita et al., 2019; Xia et al., 2022) attempts to tackle this intractability by replacing the KL-divergence term with distribution-independent surrogates. A widely used surrogate (Voita et al., 2019) is $\lambda \sum_{z_i \in \mathbf{z}} Pr[z_i \neq 0]$, which can be seen as a special case of the KL-divergence term that assumes a constant ratio $\log \frac{q_\Phi(z_i)}{p(z_i|D)} = \lambda$. Though this surrogate circumvents the intractability issue, it is often challenging to identify the right $\lambda$ for a given sparsity target $s$ (Li et al., 2021) . Other work (Xia et al., 2022) utilizes surrogates in the form of Lagrangian Multipliers (Wang et al., 2020c) to enforce *sparsity in expectation* for a given target. Though this approach is able to achieve target sparsities in a probabilistic manner, its performance highly relies on the gate thresholds and may lead to limited subnetwork performance, as illustrated in Figures 1b and 1c.

In light of the limitations of previous work, we introduce the notion of *almost-sure* sparsity and propose a novel surrogate which allows us to learn empirically good approximate posteriors as well as discover subnetworks with desired target sparsities *almost surely*. The intuition behind the *almost-sure* sparsity is straightforward. Note that a model has sparsity $s$ provided a fraction $s$ of the gates are closed in the network. From a probabilistic perspective, it is natural to ask a subnetwork to be "confident" about which gates should

---

[1]The posterior distribution $p$ also depends on the models but we ignore it here since it does not change the inequality.

be closed. In other words, gates should be closed with high probability. Mathematically, an event is said to happen almost surely, if it happens with probability 1 (Jacod & Protter, 2003). Formally, we define *almost-sure* sparsity as follows.

**Definition 1 (Almost-sure Sparsity)** *Given $s \in (0,1)$, gate variables $\mathbf{z}$ have almost-sure sparsity $s$ if $\exists \mathbf{z}_{close}, \mathbf{z}_{open} \subseteq \mathbf{z}$, such that $Pr[z_i = 0] = 1$, $\forall z_i \in \mathbf{z}_{close}$ and $Pr[z_i = 1] = 1$, $\forall z_i \in \mathbf{z}_{open}$, where $\mathbf{z}_{close} \cap \mathbf{z}_{open} = \emptyset$, $\mathbf{z}_{close} \cup \mathbf{z}_{open} = \mathbf{z}$, and $|\mathbf{z}_{close}| = s|\mathbf{z}|$.*

We argue that the *almost-sure* sparsity is better aligned with the sparsity notion we need in static subnetworks and enables the subnetwork discovery with desired sparsity targets. Next, we present learning objectives designed to achieve *almost-sure* sparsity targets specified by users.

## 3.2 Learning Objective with Almost-sure Sparsity

We aim to learn a good approximate posterior $q(\mathbf{z}; \Phi)$ with desired almost-sure sparsity. In this paper, we adopt the Hard Concrete distribution (Louizos et al., 2018) as the basic form of the approximate posterior $q(\mathbf{z}; \Phi)$, given its continuous-discrete nature and its wide application in model pruning (Voita et al., 2019; Xia et al., 2022).

Hard Concrete distribution has its support over the closed interval $[0, 1]$ and non-zero probability mass at 0 and 1. Hard Concrete distribution is derived by stretching and collapsing the Concrete distribution (Maddison et al., 2016), as illustrated in Figure 3a. We introduce derivation details in Appendix A.2. For each gate $z_i \in [0, 1]$ following Hard Concrete distribution, the corresponding probability mass at 0 and 1 with respect to $q(z_i; \phi_i)$ are given as $q(z_i = 0; \phi_i) =$

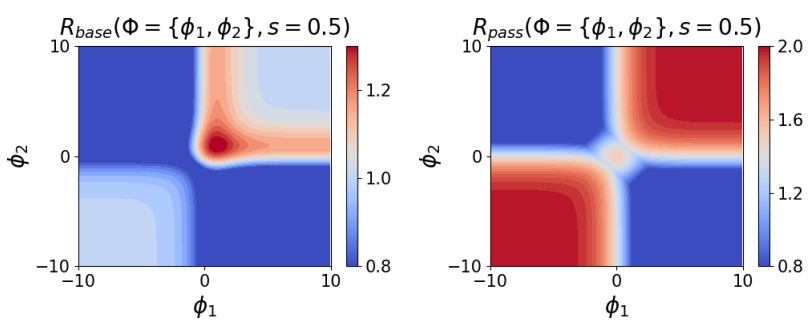

Figure 2: Values of $\mathcal{R}_{base}$ and $\mathcal{R}_{pass}$ with $\Phi = \{\phi_1, \phi_2\}$ and $s = 0.5$.

$\text{sigmoid}\left(\beta \log\left(\frac{-\gamma}{\zeta}\right) - \phi_i\right)$, $q(z_i = 1; \phi_i) = \text{sigmoid}\left(\phi_i - \beta \log\left(\frac{1-\gamma}{\zeta-1}\right)\right)$. For simplicity of notation, we denote $q_0(\phi_i) := q(z_i = 0; \phi_i)$, the gate *closing* probability, and $q_1(\phi_i) := q(z_i = 1; \phi_i)$, the gate *opening* probability. Due to the monotonicity of the sigmoid function, when $\phi_i$ increases, $q_1(\phi_i)$ increases and $q_0(\phi_i)$ decreases, and gate $z_i$ is more likely to open. We further define $q_{nb}(\phi_i) = 1 - q_0(\phi_i) - q_1(\phi_i)$ as the probability for $z_i$ being *non-binary*. We use $\beta = 0.33$, $\gamma = -0.1$, and $\zeta = 1.1$ by default, following previous work (Voita et al., 2019). Clearly, the closing and opening probability of each $z_i \in \mathbf{z}$ are differentiable functions of $\phi_i \in \Phi$, as shown in Figure 3b. By jointly learning $\Phi$ with the network parameters, we are able to almost-surely close (resp. open) gates $z_i \in \mathbf{z}$ by continuously increasing (resp. decreasing) the values of $\phi_i \in \Phi$, using gradient-descent optimizers (e.g., Adam (Kingma & Ba, 2015)). At each training iteration, gates are sampled w.r.t. the learnt distribution and then applied to attention heads to achieve pruning.

At the end of pruning, we want $q(\mathbf{z}; \Phi)$ to achieve almost-sure sparsity for a given target $s$. Our strategy is to design a learning objective that meets the desired almost-sure sparsity at its optimum, and optimize it along with model training. It is worth pointing out that there exists a family of learning objectives satisfying this criterion. However, not all of them can be easily optimized to their minimum, especially by gradient descent optimizers (Kingma & Ba, 2015). For example, one may propose to minimize the following objective.

$$\mathcal{R}_{base}(\Phi, s) = \sum_{i=1}^{|\theta|} q_{nb}(\phi_i) + \left| s|\theta| - \sum_{i=1}^{|\theta|} q_0(\phi_i) \right| \tag{6}$$

It can be easily seen that $\mathcal{R}_{base}$ takes on its minimum value 0 when achieving almost-sure sparsity $s$. However, there exist local optima that may prevent gradient descent optimizers from converging to the global optimum. To illustrate this, for simplicity, we visualize the values of $\mathcal{R}_{base}$ in a 2-gates setting $\mathbf{z} = \{z_1, z_2\}$ in Figure 2.

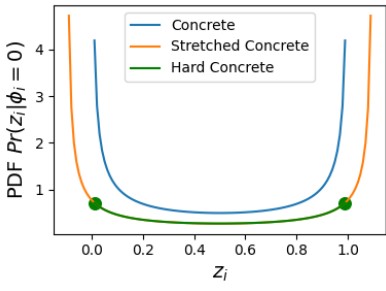
(a) Hard concrete distribution.

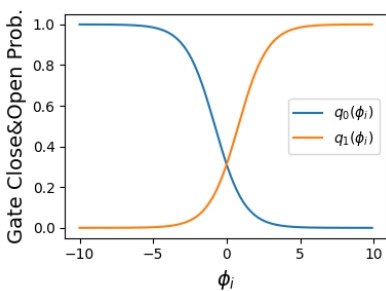
(b) Closing and opening probabilities.

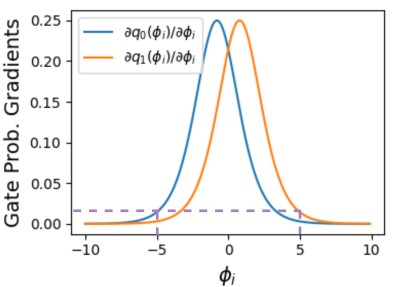
(c) Gate probability gradients.

Figure 3: (a) Hard concrete distribution derived by streching-and-collapsing a Concrete distribution. (b) Closing and opening probability of gating variables are differentiable functions of $\phi_i$. (c) The gradient values of both gate closing and opening probabilities quickly approach 0 as $\phi_i$ increases or decreases. Hard concrete distribution is parameterized with $\beta = 0.33$, $\gamma = -0.1$, $\zeta = 1.1$, following previous work (Voita et al., 2019).

With 2 gates and a sparsity target $s = 0.5$, we want one gate to be almost-surely closed and the other gate almost-surely opened. In Figure 2, such global optima correspond to the top-left and bottom-right corner where one of $\phi_1$ or $\phi_2$ takes on a high value and the other takes on a low value. However, it can be clearly observed that there exist a local optimum in the top-right region which corresponds to the situation where both gates are open with high probability. In other words, with $\mathcal{R}_{base}$, if both $\phi_1$ and $\phi_2$ happen to take positive values due to noise from the training process or bad initialization, the gradient descent direction will increase the probability for both gates to be open and fail to meet the sparsity target $s = 0.5$ by delivering overly dense models. Under weak conditions[2], we can prove that the gradient descent direction of $\mathcal{R}_{base}$ always leads to a higher opening probability for gate $z_i$ if $\phi_i \geq \log(\frac{-1-\sqrt{1-g(a)g(-a)}}{g(a)})$, where $g(a) = 2e^a - e^{-a}$, $a = \beta \log(\frac{-\gamma}{\zeta})$. The proof is presented in Appendix A.3.

In light of the limitation of $\mathcal{R}_{base}$, we propose the following learning objective,

$$\mathcal{R}_{pass}(\Phi, s) = \sum_{i=1}^{|\theta|} q_{nb}(\phi_i) + \left| s|\theta| - \sum_{i=1}^{|\theta|} q_0(\phi_i) \right| + \left| (1-s)|\theta| - \sum_{i=1}^{|\theta|} q_1(\phi_i) \right| \tag{7}$$

$\mathcal{R}_{pass}$ does not suffer from the local optimality issue that $\mathcal{R}_{base}$ does, as shown in Figure 2. In fact, we can show that minimizing $\mathcal{R}_{pass}$ always generates neither over-sparse nor over-dense subnetworks. In order to show this formally, we define the expected number of opened gates ($\frac{1}{|\theta|}\sum_{i=1}^{|\theta|} q_1(\phi_i)$) as the *expected density*, and the expected number of closed gates ($\frac{1}{|\theta|}\sum_{i=1}^{|\theta|} q_0(\phi_i)$) as the *expected sparsity*. We have the following lemma.

**Lemma 1** *Minimizing $\mathcal{R}_{pass}$ always generates sparse subnetworks whose expected sparsities are no more than $s$ and expected densities are no more than $1 - s$, for any given sparsity target $s \in (0, 1)$.*

Proof can be found in Appendix A.4. By substituting the KL-divergence term in Eq. 5 with $\mathcal{R}_{pass}$, we obtain the PASS optimization objective where $\lambda$ is the regularization coefficient.

$$\mathcal{L}_{pass}(\theta, \Phi) = -\mathbb{E}_{q(\mathbf{z};\Phi)}[\log P(D|\theta \odot \mathbf{z})] + \lambda \mathcal{R}_{pass}(\Phi, s)$$
$$\theta_{pass}, \Phi_{pass} = \underset{\theta, \Phi}{\arg\min}\ \mathcal{L}_{pass}(\theta, \Phi) \tag{8}$$

### 3.3 Concentrator

To further improve model inference efficiency, we propose the use of concentrator. Wang et al. (2020a) observed that the auxiliary operations in multi-head attention computation (e.g., reshaping and transposing

---

[2]We assume the Hard Concrete distribution is equally stretched in both directions, which gives $\gamma + \zeta = 1$.

matrices, heads splitting, and concatenation) account for 73% of the overall latency in attention layers. The run-time overhead can hardly be avoided as long as there exist unpruned heads in the attention layers. Consider subnetworks of the same attention head sparsity. Intuitively, if the unpruned attention heads are inclined to *concentrate* among a few layers, the other layers can be entirely skipped, saving the run-time overhead and improving inference efficiency. Given this, we propose the *concentrator* to encourage the unpruned attention heads to be concentrated on as few layers as possible.

Given a Transformer-based model of $L$ layers and $H$ heads per layer, the concentrator is defined as $\mathcal{R}_{conc}(\Phi) = \sum_{l=1}^{L} \left( 1 - \prod_{h=1}^{H} q_0(\phi_{l,h}) \right)$, where $\phi_{l,h}$ is the distribution parameter for the $h$-th gate variable on the $l$-th layer. Notice that $1 - \prod_{h=1}^{H} q_0(\phi_{l,h})$ indicates if the $l$-th layer can be entirely skipped: it takes on a value 0 only if all heads of the layer have a closing probability 1. $\mathcal{R}_{conc}$ is a summation of the layer-wise indicators over all layers and has regularization effects by penalizing the levels of unconcentration. We introduce $\lambda_c$ to control the concentrator effects and obtain the following optimization objective for PASSCONC, i.e., PASS with concentrator.

$$\mathcal{L}_{passconc}(\theta, \Phi) = -\mathbb{E}_{q(\mathbf{z};\Phi)}[\log P(D|\theta \odot \mathbf{z})] + \lambda \mathcal{R}_{pass}(\Phi, s) + \lambda_c \mathcal{R}_{conc}(\Phi)$$
$$\theta_{passconc}, \Phi_{passconc} = \arg\min_{\theta, \Phi} \mathcal{L}_{passconc}(\theta, \Phi) \tag{9}$$

### 3.4 Clipping and Reopening

In practice, with proper training settings, the proposed approach can discover subnetworks with the desired sparsities and high accuracy. Note that we approach almost sure sparsity by increasing or decreasing $\phi_i \in \Phi$ with gradient-descent optimizers. However, as $\phi_i$'s increase or decrease, their gradients quickly converge to 0 as illustrated in Figure 3c. Consequently, gates closed (resp. opened) with high probability in early training stage are unlikely to be self-adaptively re-opened (resp. closed) in later training iterations by gradient-descent optimizers, which may lead to sub-optimal pruning results. We propose to resolve this issue with a clipping and selective reopening strategy. The idea of clipping has been widely used in training deep learning models to avoid gradient exploding and vanishing (Zhang et al., 2019; Koloskova et al., 2023). In this same spirit, we clip $\phi_i$ to predefined ranges to alleviate the aforementioned issues caused by small gradients. In our implementation, we empirically clip all $\phi_i$'s to the range $[-5, 5]$ to avoid the vanishing of gradients to excessively small values (see Figure 3c). Randomness has been widely observed to be helpful in neural network training (Bottou, 2010). To further incentivize training dynamics, we propose to randomly reopen closed gates with respect to the gate quality. There is a line of work on how to measure gate qualities (Michel et al., 2019; Voita et al., 2019; Ding et al., 2017), among which we choose head confidence (Voita et al., 2019) in our implementation because it has been found to be an informative notion (Behnke & Heafield, 2020) and requires little to no additional computation. The confidence of a head is the average maximum attention weights of tokens in a set of sentences (Voita et al., 2019). We normalize confidence scores for each attention head and reopen almost-surely closed gates[3] with a probability equal to the normalized scores.

Notably, the clipping and reopening strategy is designed to be applied jointly. Without clipping to a proper range, an opened gate may remain fully open due to the nearly zero gradients and finally result in overly-dense subnetworks as the reopening strategy progresses. Similarly, without the randomly reopening strategy, the subnetwork may be trapped by certain local optimums and lead to suboptimal performance even with the clipping strategy.

## 4 Evaluation

### 4.1 Evaluation Setup

#### 4.1.1 Model, Data, and Metrics

**Model**: We investigate two widely-studied Transformer models: encoder-decoder (ED) Transformer (Vaswani et al., 2017) and the encoder-only Transformer, BERT (Devlin et al., 2018). We use the FAIRSEQ toolkit (Ott et al., 2019) to implement a 6-layer ED Transformer with 72 heads in total, and the HUGGING

---

[3]To reopen an almost-surely closed gate $z_i$, we manually decrease its closing probability by increasing $\phi_i$.

FACE codebase (Wolf et al., 2020) to implement a 12-layer BERT-base with 144 heads in total. We believe our setting has covered the most important attention head variants, especially that the ED Transformer includes both encoder (self-attention) and decoder (self- and cross-attention) modules, and therefore suffices a comprehensive evaluation setup. [4]

**Datasets**: Following previous work (Li et al., 2021), the ED Transformer model is trained and evaluated on the IWSLT14 German-to-English translation dataset (Cettolo et al., 2014). The BERT-base model is fine-tuned and evaluated on 4 benchmark NLP tasks from the GLUE benchmark (Wang et al., 2018) including the Multi-Genre Natural Language Inference (MNLI) dataset (Williams et al., 2018), the Question-answering NLI (QNLI) dataset (Rajpurkar et al., 2016), the Quora Question Pairs (QQP) dataset (Sharma et al., 2019), and the Stanford Sentiment Treebank (SST-2) dataset (Socher et al., 2013). We choose these 4 datasets of the largest training splits from the GLUE benchmark to ensure that gate variables can be sufficiently trained to convergence.

**Metrics**: We use BLEU score (Papineni et al., 2002) as the metric to measure model performance on the translation task following previous work (Li et al., 2021; Michel et al., 2019), and use accuracy as the metric on the GLUE benchmark tasks following Wang et al. (2018). In addition, we are also interested in the efficiency improvements achieved by PASS and PASSCONC. We use wall clock time to measure the efficiency w.r.t. latency.

### 4.1.2 Baselines

We consider three strong baselines that prune attention heads to a specified sparsity level.

**Differentiable Subset Pruning (DSP)**. DSP (Li et al., 2021) applies the Gumbel-softmax trick (Gumbel, 1954) to select the top-$K$ attention heads for a given sparsity target. DSP learns a $K$-hot vector $g_h$ by iteratively applying Gumbel-softmax $K$ times, where $g_h = \sum_{k=1}^{K} g_h^k = \sum_{k=1}^{K} \frac{exp(r_h^k/\tau)}{\sum_{h'=1}^{H} exp(r_{h'}^k/\tau)}$, $r_h^k = r_h^{k-1} + \log(1 - g_h^{k-1})$, and $r_h^1 = w_h + n_h$. $w_h$ denotes trainable parameter indicating head importance, $n_h \sim Gumbel(0, 1)$ is Gumbel noise, and $\tau$ is a hyper-parameter that controls the annealing temperature.

**Lagrangian Multiplier (LAG)**. A recent line of work (Xia et al., 2022; Wang et al., 2020c) employs Lagrangian Multiplier (Wang et al., 2020c) to enforce *sparsity in expectation*. Given a sparsity target $s$, LAG trains models along with the regularization term $\mathcal{R}_{lag} = \lambda_1(\hat{s} - s) + \lambda_2(\hat{s} - s)^2$, where $\hat{s}$ is the *expected sparsity*. $\lambda_1$ and $\lambda_2$ are trainable parameters and will be optimized jointly in training.

**Voita et al. (2019) (Voita)**. Voita et al. (2019) prunes attention heads by applying the stochastic approximation to $L_0$ regularization (Louizos et al., 2018) to gate closing probabilities. Voita et al. (2019) achieves pruning by jointly training models with the following regularization term $\mathcal{R}_{voita}(\Phi) = \lambda_v \sum_{h=1}^{|\mathcal{H}|} (1 - q_0(\phi_h))$, where $\lambda_v$ can be used to *indirectly* control the achieved sparsities. Following the previous work (Li et al., 2021), we use grid search to find the correct $\lambda_v$ values for each sparsity setting.

As observed in Li et al. (2021), $\lambda_v$ is in general hard to tune and there could exist certain sparsity targets that cannot be achieved by tuning $\lambda_v$, which makes Voita's approach not a good fit if the user demands a sparse Transformer model of customized number of attention heads. In our work, we propose to enforce the sparsity constraints explicitly using a novel sparsifier (see Section 3.2) and further push the envelope of efficiency by developing the concentrator (see Section 3.3). The design of our sparsifier, concentrator, as well as the clipping and reopening strategy has not been proposed in previous attention pruning literature and contributes to the novelty of our work.

We adopt the same implementation of DSP and Voita as in Li et al. (2021) [5]. We implement LAG by training models with the regularization term $\mathcal{R}_{lag} = \lambda_1(\hat{s} - s) + \lambda_2(\hat{s} - s)^2$, where $s$ is the user-specified sparsity target, $\lambda_1$ and $\lambda_2$ are trainable parameters, and $\hat{s}$ is the expected sparsity over all attention heads. Following Louizos et al. (2017), we estimate the value of each gate as $\hat{z}_i = \min\{1, \max\{0, \text{sigmoid}(\log(\phi_i)(\zeta - \gamma) + \gamma)\}\}$, for $\phi_i \in \Phi$. The expected sparsity $\hat{s}$ over attention heads $\mathcal{H}$ is computed as $\hat{s} = \frac{1}{|\mathcal{H}|} \sum_{i=1}^{|\mathcal{H}|} 1 - \hat{z}_i$.

---

[4]Observing the emergence of large language models (LLM) (Chang et al., 2024) that are typically decoder-only Transformer models, it is intriguing to evaluate the scalability of our approaches on large-scale LLMs and we leave it in our future work.

[5]https://github.com/rycolab/differentiable-subset-pruning.

Table 1: Subnetwork performance on IWSLT14 De-En translation (left) and GLUE benchmark tasks (right).

| | BLEU | | | | | Accuracy (%) | | | | |
|---|---|---|---|---|---|---|---|---|---|---|
| K | PASS | PASSCONC | DSP | LAG | Voita | PASS | PASSCONC | DSP | LAG | Voita |
| 16 | **32.73** | 32.70 | 31.40 | 30.91 | 27.55 | **86.27** | 85.25 | 84.47 | 83.84 | 84.83 |
| 32 | 33.45 | **33.48** | 33.42 | 32.66 | 32.80 | 87.59 | 86.47 | **88.36** | 86.99 | 87.15 |
| 48 | 33.89 | 33.91 | **34.00** | 33.12 | 32.97 | **88.65** | 88.30 | 88.52 | 88.02 | 88.02 |
| 64 | 34.01 | **34.05** | 33.89 | 33.02 | 33.20 | 88.72 | 88.62 | **88.81** | 88.40 | 84.20 |

### 4.1.3 Protocols

We express sparsity targets over attention heads $\mathcal{H}$ interchangeably as $s \in (0,1)$ and as integer $K$ where $K = \lfloor (1-s)|\mathcal{H}| \rfloor$, the number of unpruned heads. Unless stated otherwise, for a given sparsity target $K$, we evaluate all methods by selecting the top-$K$ most important heads w.r.t. the corresponding ranking metrics, i.e., the gate opening probabilities for PASS, PASSCONC, Voita, and LAG, and the head importance score $w_h$ for DSP. Detailed hyper-parameter settings are in Appendix B. We test all methods on both architectures with target tasks (30 training epochs for ED Transformer; 3 fine-tuning epochs for BERT-base as in Li et al. (2021)). All experiments are conducted on a high performance compute cluster equipped with NVIDIA P100 GPUs (each with 12GB GPU RAM). Codebase is available on https://github.com/DujianDing/PASS.

We evaluate the performance of PASS and PASSCONC in Section 4.2, demonstrate the speedups brought by PASSCONC in Section 4.3, validate the effectiveness of concentrator as well as the clipping and reopening strategy in Section 4.4, present the patterns of pruned attention heads in Section 4.5, and show more performance results in Appendix C.

## 4.2 PASS and PASSCONC Improve Model Performance

We investigate the model performance of subnetworks identified by PASS, PASSCONC and all baselines under various sparsity constraints. We compare all five methods on both ED Transformer and BERT-base models. The results are summarized in Table 1. Detailed performance results on each GLUE benchmark task are in Appendix C.

On IWSLT14 German-to-English translation task, PASS and PASSCONC outperform all 3 baselines in a majority of sparsity settings. When $K = 16$, both PASS and PASSCONC achieve BLEU scores of 32.7, which is 1.3 higher than DSP, 1.8 higher than LAG, and 5.2 higher than Voita. On the GLUE benchmark tasks, we observe a similar trend in high sparsity situations. When $K = 16$, PASS and PASSCONC achieve average model accuracy of 86.27% and 85.25% respectively, while DSP drops to 84.47%, LAG drops to 83.84%, and Voita drops to 84.83%. When sparsity targets are low, PASS is able to match or outperform all 3 baselines, while PASSCONC can be outperformed by the strongest baseline DSP while still being comparable to the remaining two.

One interesting observation is that Voita delivers surprisingly low accuracy in low sparsity settings (e.g., $K = 64$) with GLUE benchmark tasks. The degraded performance can be attributed to its intrinsic sensitivity to the choice of $\lambda_v$, which is used to indirectly control sparsity targets. Li et al. (2021) observed that a small increase in $\lambda_v$ (e.g., $0.0009 \rightarrow 0.0014$) may lead to drastic change of achieved sparsity (e.g., the number of unpruned heads decreases from 30 to 11), which suggests that Voita is inadequate when users require subnetworks of pre-defined number of attention heads.

## 4.3 PASSCONC Improves Model Efficiency

We evaluate the attention speedups for subnetworks identified under various sparsity constraints, at inference time. We report the inference speedups in comparison to the unpruned model. The results are summarized in Figure 4 and Table 2. Detailed results on each GLUE benchmark task can be found in Appendix C.

On the GLUE benchmark tasks with BERT-base models, PASSCONC outperforms all baselines across a majority of sparsity constraints with great efficiency improvements and comparable or better accuracy (see

Table 2: Attention speedups on IWSLT14 De-En translation (left) and GLUE benchmark tasks (right).

| | Speedup (%) | | | | | Speedup (%) | | | | |
|---|---|---|---|---|---|---|---|---|---|---|
| K | PASS | PASSCONC | DSP | LAG | Voita | PASS | PASSCONC | DSP | LAG | Voita |
| 16 | 144.3 | **162.8** | 141.1 | 141.1 | 142.7 | 114.4 | **185.2** | 123.1 | 120.4 | 126.1 |
| 32 | 115.5 | **118.7** | **118.7** | 110.4 | 117.6 | 107.1 | **135.6** | 107.3 | 105.4 | 105.3 |
| 48 | 101.6 | 104.1 | **105.8** | 102.4 | 102.4 | 103.1 | **109.3** | 102.7 | 102.9 | 103.8 |
| 64 | 100.8 | **104.1** | 100.0 | 100.0 | 100.0 | 103.2 | **106.4** | 102.9 | 103.0 | 103.0 |

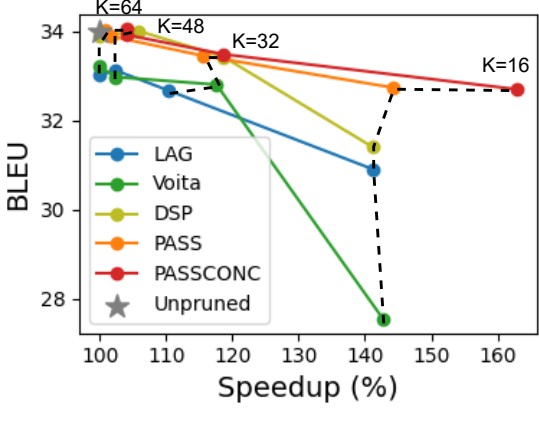

(a) IWSLT14 De-En translation.

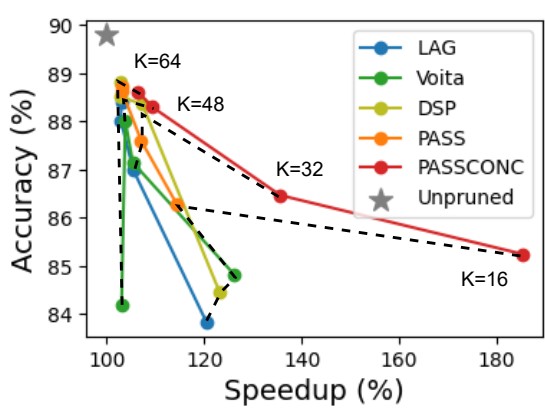

(b) GLUE benchmark tasks.

Figure 4: Subnetwork performance v.s. attention speedups on IWSLT14 De-En translation task and GLUE benchmark tasks. We add dashed lines (−−) to connect data points with the same sparsity constraints for better comparison. The unpruned model performance is included for reference purpose.

Figure 4). When $K = 16$, PASSCONC achieves a 185.2% speedup, which is 60% higher than all baselines, and an average accuracy 85.25% that is also higher than DSP, LAG, and Voita. PASS has a better accuracy but a lower speedup. As the sparsity targets decrease (i.e, as $K$ increases), the speedups achieved by all methods in general goes down but PASSCONC always dominates the competition in terms of efficiency, at the price of a relatively small drop in performance. On IWSLT14 German-to-English task with ED Transformer model, PASSCONC outperforms all baseline methods in almost all sparsity settings (see Table 2). When $K = 16$, PASSCONC achieves a 162.8% speedup, which is more than 20% higher than all baselines, with at least 1.3 higher BLEU scores.

## 4.4 Ablation Study

Previous analysis of PASS and PASSCONC in Section 4.3 demonstrates the significant efficiency improvements brought about by the concentrator (see Section 3.3). In this section, we validate that (1) by introducing the concentrator loss, PASSCONC is able to prune more attention layers entirely (Figure 5a), and (2) the clipping and reopening strategy is necessary for PASSCONC to obtain significant efficiency improvements (Figure 5b). We report results using BERT-base models and the GLUE benchmark tasks. In Figure 5a, we observe that PASSCONC with concentrator is able to prune up to 7 out of 12 attention layers with high sparsity targets, which is 3 layers more than not using concentrator and therefore validates its effectiveness in pruning redundant attention layers entirely to bring significant speedups. In Figure 5b, we observe that without the clipping and reopening strategy, the speedups achieved by PASSCONC can reduce by up to 70%! This observation demonstrates the necessity of dynamically re-activating closed gates to help the model converge to cost-effective regions, as desired by the concentrator.

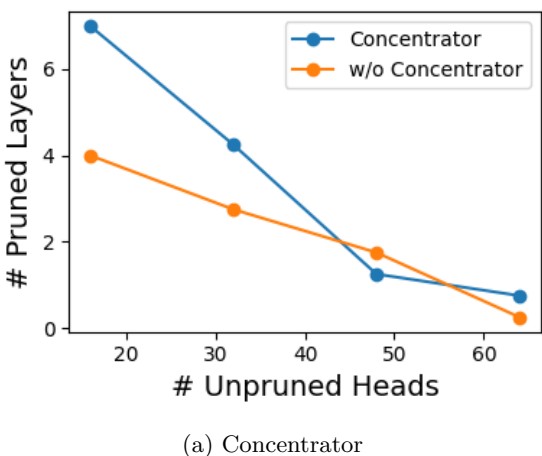

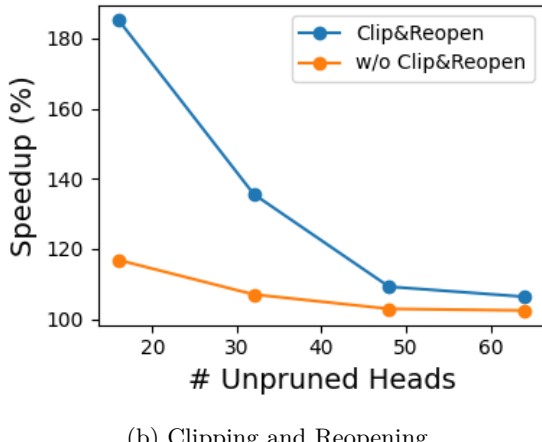

(a) Concentrator

(b) Clipping and Reopening

Figure 5: (a) Concentrator is critical for PASSCONC to prune more attention layers entirely. (b) The clipping and reopening strategy is necessary for PASSCONC to obtain significant efficiency improvements. All results are reported with BERT-base models on the GLUE benchmark tasks.

To disentangle the benefit of the clipping and reopening strategy from the proposed sparsity objective, we implement the Voita approach with the clipping and reopening strategy and compare it with PASS and PASSCONC on the SST-2 dataset from the GLUE benchmark. Results are summarized in Figure 6. By clipping and reopening gates as Voita prunes attention heads, the accuracy of the resulting subnetworks drastically drops to 50.9% when $K = 16$ while PASS and PASSCONC achieve accuracy at least 89.3%. One possible reason is that, the clipping and reopening strategy reopens fully closed gates along the pruning process which tends to continue reducing the subnetwork sparsity levels. Without our proposed sparsity objective, Voita fails to converge to the desired sparsity levels and leads to decayed performance when we extract the sparse subnetworks according to the user-specified sparsity targets.

## 4.5 Distribution of Heads

We visualize the distribution of unpruned heads to better understand the dynamics of the pruning process (see Figure 7). We report the frequency of heads unpruned by PASSCONC, averaged over all datasets (IWSLT14 De-En translation task for ED-Transformer and GLUE benchmark tasks for BERT-base) and sparsity settings (K=16, 32, 48, 64). In each layer, the heads are sorted by the unpruned frequency for clearer presentation. We also report the average number of unpruned heads per layer for both ED-Transformer and BERT-base to provide a quantitative comparison (see Tables 3 and 4). For ED Transformer (Figure 7a and table 3), we observe that the encoder self-attention heads are typically less important than cross-attention and decoder self-attention heads at test time, and therefore are pruned more frequently (the average number of unpruned encoder self-attention heads are at least 5 fewer than the other two attention types). This observation suggests the high level of redundancy in encoder attention heads from properly trained Enc-Dec Transformers, which conforms with the observations in

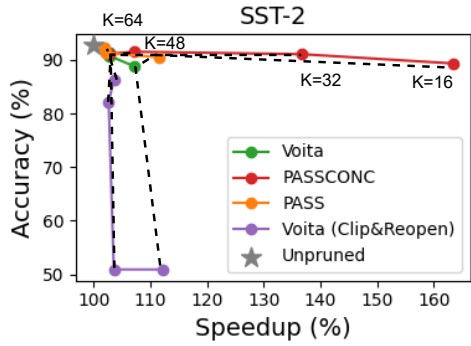

Figure 6: Voita augmented with clipping and reopening strategy leads to decayed subnetwork performance.

Li et al. (2021) and Voita et al. (2019). For BERT-base (Figure 7b and table 4), we observe that the middle layers (Layer 2,3,4,6) are likely to contain more unpruned heads, in comparison to the bottom (Layer 1) and

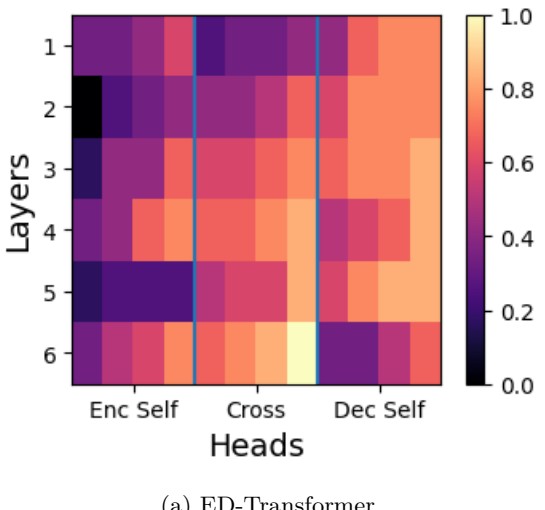

(a) ED-Transformer

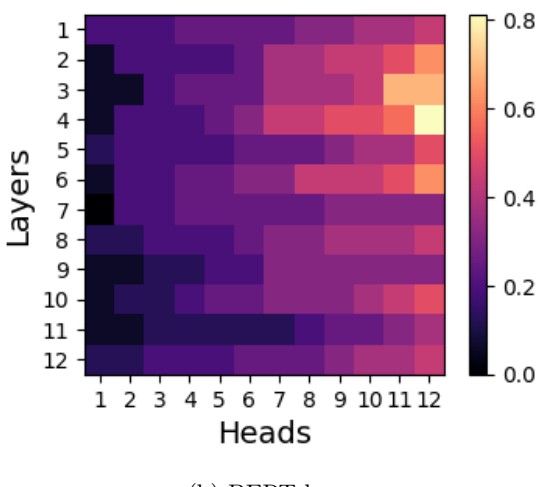

(b) BERT-base

Figure 7: Distribution of unpruned heads across layers. Different colors represent the unpruned frequency for each head across all datasets (IWSLT14 De-En translation task for ED-Transformer and GLUE benchmark tasks for BERT-base) and sparsity settings (K=16, 32, 48, 64).

Table 3: The average number of unpruned heads per layer in ED-Transformer on IWSLT14 De-En translation task across sparsity settings (K=16, 32, 48, 64).

| Layer | 1 | 2 | 3 | 4 | 5 | 6 | SUM |
|---|---|---|---|---|---|---|---|
| Enc Self | 1.67 | 1.00 | 1.67 | 2.17 | 0.92 | 2.17 | 9.58 |
| Cross | 1.33 | 2.00 | 2.58 | 2.92 | 2.50 | 3.25 | 14.58 |
| Dec Self | 2.58 | 2.83 | 3.00 | 2.58 | 3.00 | 1.83 | 15.83 |

top (Layer 7-12) layers, which is also aligned with the observation in previous work (Prasanna et al., 2020b; Sajjad et al., 2023).

## 4.6 Overhead Analysis

To realize the efficiency improvements achieved by attention head pruning, we index the unpruned attention heads and only perform matmul on the indexed tensors during inference. In Sections 4.2 to 4.4, we implement all approaches with indexing and evaluate the speedups w.r.t. the unpruned model using indexing to better understand the efficiency improvements solely caused by different attention sparsities. In this subsection, we investigate the overhead of indexing by comparing our methods with indexing to the unpruned model without indexing (matmul is performed on the original unpruned matrix), using a 72-head Encoder-Decoder Transformer on 2 Intel Broadwell CPUs @ 2.2GHz, and 1 NVIDIA P100 Pascal GPU. Comparison results are summarized in Table 5, where we report the average latency achieved by subnetworks of K unpruned heads on CPU and GPU devices separately.

Table 4: The average number of unpruned heads per layer in BERT-base on GLUE benchmark tasks across sparsity settings (K=16, 32, 48, 64).

| Layer | 1 | 2 | 3 | 4 | 5 | 6 | 7 | 8 | 9 | 10 | 11 | 12 |
|---|---|---|---|---|---|---|---|---|---|---|---|---|
| Enc Self | 3.38 | 3.81 | 4.00 | 4.44 | 3.19 | 4.00 | 2.88 | 3.25 | 2.63 | 3.25 | 2.13 | 3.06 |

Table 5: The average inference latency for subnetworks extracted by indexing unpruned heads, compared to the full model latency without indexing. On CPU-only devices, the subnetworks can achieve speed-ups up to 48 heads unpruned out of 72.

| K | 1 | 2 | 4 | 8 | 16 | 32 | 48 | 64 | No Indexing |
|---|---|---|---|---|----|----|----|----|-------------|
| GPU (1e-4 s) | 1.6 | 1.6 | 3.2 | 4.6 | 8 | 9.1 | 10.8 | 11 | 4.3 |
| CPU (1e-4 s) | 16.9 | 16.2 | 27.5 | 44.3 | 68.1 | 82.2 | 95.5 | 102.9 | 96 |

Intuitively, on CPU-only devices where matmul operation is dominantly time-consuming, the overhead of indexing could be negligible and the subnetworks can achieve speed-ups up to 48 heads unpruned out of 72. In contrast, on devices specialized for in-parallel matmul operations such as GPUs, the indexing approach may cause non-negligible overheads and head pruning achieves speed-ups only with high-sparsity targets, such as when less than 8 heads are retained from the subnetworks, as shown in Table 5.

It is worth noting that, indexing in our work is used to reflect the attention sparsities and is only necessary when we assume no specialized software supports. With software supports such as high-performance sparse matrix multiplication libraries [6], the indexing operation is no longer necessary and can be completely removed to avoid overheads.

## 5 Related Work

*Unstructured pruning* has been well studied in the literature (Gupta & Agrawal, 2022) and dates back to Optimal Brain Damage (LeCun et al., 1989). Unstructured pruning prunes individual parameters and identifies subnetworks of high sparsity. However, unstructured pruning hardly achieves practical efficiency improvements without specialized software and hardware support (Xia et al., 2022). In contrast, *structured pruning* prunes groups of parameters within certain structures (e.g., channels and attention heads). Structured pruning has been widely explored in computer vision tasks (He et al., 2017) and has started to attract research interest in the NLP community. Research efforts have been devoted to designing pruning strategies at both coarse- and fine-grained levels (Xia et al., 2022; Prasanna et al., 2020a) over structures like feed-forward layers and attention heads. Previous work on attention head pruning (Li et al., 2021; Michel et al., 2019; Voita et al., 2019) either highly relies on threshold tuning or enforces hard structural constraints on model structure in the early training stages, which could lead to limited subnetwork performance. We focus on structured pruning and propose the notion of *almost-sure* sparsity to overcome the above limitations.

In addition to pruning, many other techniques have been developed to obtain inference efficiency for deep learning models. Other than sparsity over the number of attention heads, a line of work (Roy et al., 2021; Child et al., 2019) focuses on sparsifying the attention distribution over tokens for each head to improve efficiency and head diversity. Recently, Correia et al. (2019) and Treviso et al. (2021) propose to adaptively sparsify the attention distribution by enforcing low attention scores to be exactly 0 through $\alpha$-entmax (Peters et al., 2019). Linformer (Wang et al., 2020b) develops efficient self-attention design of linear complexity by using low-rank matrix approximation. FlashAttention (Dao et al., 2022; Dao, 2023) focuses on I/O access in attention computation and achieves linear memory complexity by leveraging the hardware memory hierarchy and minimizing unnecessary data transfers. Other techniques include quantization (Shen et al., 2020), knowledge distillation (Hinton et al., 2015), parameter sharing (Ling et al., 2015), tensor decomposition (Oseledets, 2011), neural architecture search (Zhang et al., 2024; Zheng et al., 2023) and more. We refer interested readers to (Menghani, 2023; Gupta & Agrawal, 2022; Treviso et al., 2022) for a comprehensive survey.

## 6 Discussion and Conclusion

We propose a novel notion of *almost-sure* sparsity, develop a generic framework for **P**runing with **A**lmost-**S**ure **S**parsity (PASS) targets, and demonstrate its pruning capacity with attention heads. To further push

---

[6]https://docs.nvidia.com/cuda/cusparselt/

the envelope on inference efficiency, we propose a novel technique, concentrator, based on which we develop PASSCONC (**PASS** with **CONC**entrator). We also present a simple-yet-effective strategy to improve subnetwork performance by clipping and selectively reopening learned gates. We investigate PASS and PASSCONC on two widely studied Transformer models: encoder-decoder (ED) Transformer and encoder-only Transformer, BERT-base. Experiments on IWSLT14 German-to-English translation (Cettolo et al., 2014) and GLUE benchmark tasks (Wang et al., 2018) demonstrate that PASS and PASSCONC outperform the SOTA methods by identifying subnetworks of up to 1.33 higher BLEU scores, 1.44% higher accuracy, and 60% higher speedups, at the same sparsity levels.

We conclude that PASS and PASSCONC can be used to identify high performance subnetworks and help address the challenge of deploying Transformer models in resource-limited applications. We identify several important extensions for future work: (1) **More pruning structures and metrics.** We would like to explore the possibility of extending the proposed framework to multiple model structures (e.g., feed-forward layers) and prune for meeting other target footprint metrics such as latency and memory which are more accessible to users in real-world applications, in addition to sparsity. (2) **Combination with other techniques.** Since both PASS and PASSCONC are agnostic to the underlying self-attention implementation, it is intriguing to investigate the compound efficiency improvements achieved by combining our approaches with other efficiency improvement techniques such as flash attention (Dao et al., 2022; Dao, 2023) and quantization (Shen et al., 2020; Frantar et al., 2022). For example, a naive combination strategy is to first apply our attention pruning techniques to reduce the redundant heads and then leverage quantization or flash attention techniques to accelerate the computation of those unpruned heads. In the future, we would like to explore other non-trivial combination strategies to further push the envelope on efficiency gains. (3) **Large Language Models (LLMs) pruning.** LLMs are typically large-scale decoder-only Transformer models (Chang et al., 2024). Recent work (Xia et al., 2023) shows that LLMs possess a high level of parameter redundancy which can benefit from model pruning to improve inference efficiency while maintaining high performance. Though we have demonstrated the effectiveness of our approaches on both encoder and decoder modules in standard Transformer models, it is still intriguing to test the scalability of our approaches on large scale Transformer models such as LLMs. On the one hand, it is worth noting that our sparsity convergence analysis is agnostic to the model size and attention types (see Lemma 1). For decoder-only LLMs, we believe our approach will be able to prune subnetworks to desired sparsity levels. On the other hand, we acknowledge that currently the intrinsic mechanism behind the impressive emergent abilities of LLMs (Wei et al., 2022) remains unclear. How to effectively and efficiently prune attention heads from LLMs brings new challenges and opportunities which we plan to investigate in our future work.

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

# A  Proofs and Analysis

## A.1  Derivation of the Upper Bound in Eq. 4

Here we present how to derive the upper bound in Eq. 4.

$$
\begin{aligned}
& -\log \mathbb{E}_{p(\mathbf{z}|D)}[P(D|\theta \odot \mathbf{z})] \\
&= -\log \int_{\mathbf{z}} p(\mathbf{z}|D) P(D|\theta \odot \mathbf{z}) \\
&= -\log \int_{\mathbf{z}} p(\mathbf{z}|D) P(D|\theta \odot \mathbf{z}) \frac{q_{\Phi}(\mathbf{z})}{q_{\Phi}(\mathbf{z})} \\
&= -\log \mathbb{E}_{q_{\Phi}(\mathbf{z})} \left[ P(D|\theta \odot \mathbf{z}) \frac{p(\mathbf{z}|D)}{q_{\Phi}(\mathbf{z})} \right] \\
&\leq -\mathbb{E}_{q_{\Phi}(\mathbf{z})} \left[ \log \left( P(D|\theta \odot \mathbf{z}) \frac{p(\mathbf{z}|D)}{q_{\Phi}(\mathbf{z})} \right) \right] \\
&= -\mathbb{E}_{q_{\Phi}(\mathbf{z})}[\log P(D|\theta \odot \mathbf{z})] + KL\left( q_{\Phi}(\mathbf{z}) \,||\, p(\mathbf{z}|D) \right)
\end{aligned}
\tag{10}
$$

## A.2  Hard Concrete Distribution (Louizos et al., 2018)

Hard Concrete distribution is derived from the Concrete distribution (Maddison et al., 2016). The cumulative distribution function of a binary Conrete random variable $z_i^c \in (0,1)$ is as follows:

$$
Q_c(z_i^c; \beta, \phi_i) = \text{sigmoid}\left( (\log z_i^c - \log(1 - z_i^c)) \beta - \phi_i \right)
\tag{11}
$$

where $\text{sigmoid}(x) = \frac{1}{1+e^{-x}}$ is the sigmoid function and $0 < \beta < 1$ is a constant. We can obtain the Hard Concrete distribution by first stretching the support of the Concrete distribution from $(0,1)$ to $(\gamma, \zeta)$, where $\gamma < 0, \zeta > 1$, and then collapsing the probability mass on $(\gamma, 0]$ (resp. $[1, \zeta)$) to the endpoint 0 (resp. 1).

Equivalently, for each Concrete random variable $z_i^c \in (0, 1)$, we can obtain a Hard Concrete random variable $z_i = \min(1, \max(0, z_i^c(\zeta - \gamma) + \gamma)$. For each $z_i$, the closing probability $q(z_i = 0; \phi_i) = Pr[z_i^c \leq \frac{-\gamma}{\zeta-\gamma}] = Q_c(\frac{-\gamma}{\zeta-\gamma}; \beta, \phi_i)$, and the opening probability $q(z_i = 1; \phi_i) = Pr[z_i^c \geq \frac{1-\gamma}{\zeta-\gamma}] = 1 - Q_c(\frac{1-\gamma}{\zeta-\gamma}; \beta, \phi_i)$. By plugging in Eq. 11, we have the gate closing and opening probabilities.

### A.3 $\mathcal{R}_{base}$ **Local Optimum Analysis**

We show that the gradient descent direction of $\mathcal{R}_{base}$ always leads to a higher opening probability for gate $z_i$ if $\phi_i \geq \log(\frac{-1-\sqrt{1-g(a)g(-a)}}{g(a)})$, where $g(a) = 2e^a - e^{-a}$, $a = \beta \log(\frac{-\gamma}{\zeta})$. We assume that the Hard Concrete distribution is equally stretched in both directions, which gives $\gamma + \zeta = 1$ and has been widely adopted in previous work (Voita et al., 2019; Xia et al., 2022).

Without loss of generality, we focus on the high sparsity target situation ($s|\theta| \geq \sum_{i=1}^{|\theta|} q_0(\phi_i)$) and ignore the sparsity constraint $s$ in following computation because it does not change the equation. We first compute the derivative of $\mathcal{R}_{base}(\Phi)$ with respect to $\phi_i$,

$$\frac{\partial \mathcal{R}_{base}(\Phi)}{\partial \phi_i} = \lambda \left( 2sig(-\phi_i + a)sig(\phi_i - a) \right.$$
$$\left. - sig(-\phi_i - a)sig(\phi_i + a) \right) \tag{12}$$

where $\lambda > 0$, $a = \beta \log\left(\frac{-\gamma}{\zeta}\right)$ and $\beta > 0$, $\zeta > 1$, $\gamma + \zeta = 1$. In order to compute the domain for $\frac{\partial \mathcal{R}_{base}(\Phi)}{\partial \phi_i} \leq 0$, we can equally solve the following inequalities,

$$\frac{\partial \mathcal{R}_{base}(\Phi)}{\partial \phi_i} \leq 0$$
$$\iff 2sig(-\phi_i + a)sig(\phi_i - a) \leq$$
$$sig(-\phi_i - a)sig(\phi_i + a)$$
$$\iff 2\frac{sig(-\phi_i + a)}{sig(-\phi_i - a)} \leq \frac{sig(\phi_i + a)}{sig(\phi_i - a)} \tag{13}$$

where the last step is due to the non-negativity of the sigmoid function. By denoting $f_a(\phi_i) = \frac{sig(\phi_i+a)}{sig(\phi_i-a)}$, we can simplify the inequality to be $2f_a(-\phi_i) \leq f_a(\phi_i)$. The derivative of $f_a(\phi_i)$ with respect to $\phi_i$ is as follows,

$$\frac{\partial f_a(\phi_i)}{\partial \phi_i} = \frac{sig(\phi_i + a)}{sig(\phi_i - a)} \left( sig(-\phi_i - a) - sig(-\phi_i + a) \right) \tag{14}$$

Because $a = \beta \log\left(\frac{-\gamma}{\zeta}\right) = \beta \log\left(\frac{\zeta-1}{\zeta}\right) < 0$, we have $-\phi_i - a > -\phi_i + a$ and hence $sig(-\phi_i - a) > sig(-\phi_i + a)$ due to the fact that the sigmoid function is monotonically increasing. As a result, the derivative of $f_a(\phi_i)$ with respect to $\phi_i$ is always positive ($\frac{\partial f_a(\phi_i)}{\partial \phi_i} > 0$) and therefore $f_a(\phi_i)$ is monotonically increasing while $f_a(-\phi_i)$ monotonically decreases as a function of $\phi_i$.

Assuming there is $\phi_i'$ having the equality $2f_a(-\phi_i') = f_a(\phi_i')$. It is easy to show that for $\phi_i \geq \phi_i'$, the inequality $2f_a(-\phi_i) \leq f_a(\phi_i)$ holds true due to the monotonicity of $f_a(\phi_i)$ and $f_a(-\phi_i)$, and hence we have $\frac{\partial \mathcal{R}_{base}(\Phi)}{\partial \phi_i} \leq 0$ (the gradient descent direction). Notably, it indicates that, for $\phi_i \geq \phi_i'$, any gradient-descent optimizer (e.g., Adam (Kingma & Ba, 2015)) will continue increasing the value of $\phi_i$ which opens more gates and leads to undesired low sparsity, as dicussed in Section 3.2. We can derive the value of $\phi_i'$ by solving

equation $2f_a(-\phi_i') = f_a(\phi_i')$ as follows,

$$
\begin{aligned}
2f_a(-\phi_i') &= f_a(\phi_i') \\
\Leftrightarrow 2\frac{sig(-\phi_i'+a)}{sig(-\phi_i'-a)} &= \frac{sig(\phi_i'+a)}{sig(\phi_i'-a)} \\
\Leftrightarrow 2\frac{1+e^{\phi_i'+a}}{1+e^{\phi_i'-a}} &= \frac{1+e^{-\phi_i'+a}}{1+e^{-\phi_i'-a}} \\
\Leftrightarrow (2e^a - e^{-a})e^{\phi_i'} &= -2 + (e^a - 2e^{-a})e^{-\phi_i'} \\
\Leftrightarrow (2e^a - e^{-a})e^{2\phi_i'} &= -2e^{\phi_i'} + e^a - 2e^{-a}
\end{aligned}
\tag{15}
$$

which is a quadratic equation with respect to $e^{\phi_i'}$ and gives us $\phi_i' = log(\frac{-1+\sqrt{1-g(a)g(-a)}}{g(a)})$ where $g(a) = 2e^a - e^{-a}$.

## A.4   Proof of Lemma 1

We include the proof of Lemma 1 as follows.

**Proof**: We prove this lemma by showing that, in every possible case, the gradient descent direction of the objective $\mathcal{R}_{pass}$ always leads to a subregion in the search space where the expected sparsity is no more than $s$ and expected density is no more than $1 - s$. Recall that we define *expected density* as $\frac{1}{|\theta|}\sum_{i=1}^{|\theta|} q_1(\phi_i)$, and the *expected sparsity* as $\frac{1}{|\theta|}\sum_{i=1}^{|\theta|} q_0(\phi_i)$. For the simplicity of proof language, we use $E_d$ to denote the expected density and $E_s$ to denote the expected sparsity. Intuitively, as the expected density increases (resp. decreases), the expected sparsity will monotonically decrease (resp. increase). One important observation is that, *the sum of expected density and sparsity is always less than* 1, due to the fact that $q_1(\phi_i) + q_0(\phi_i) = 1 - q_{nb}(\phi_i)$ and $q_{nb}(\phi_i) > 0$.

Given a sparsity target $s$, depending on the values of expected density and sparsity, there are three possible situations: (i) the easiest case is that **the model is already neither over-sparse nor over-dense** (i.e., $E_d \leq 1 - s$ and $E_s \leq s$). In this case, we can rewrite $\mathcal{R}_{pass}(\Phi, s) = 2\sum_{i=1}^{|\theta|} q_{nb}(\phi_i)$. Minimizing $\mathcal{R}_{pass}(\Phi, s)$ amounts to minimizing $q_{nb}(\phi_i)$, which polarizes the gates to achieve the required almost-sure sparsity, until either $E_d = 1 - s, E_s < s$ or $E_d < 1 - s, E_s = s$ both of which satisfy Lemma 1. (ii) **high expected density and low expected sparsity** (i.e., $E_d \geq 1 - s$ and $E_s \leq s$), in which case we can rewrite Eq. 7 to obtain $\mathcal{R}_{pass}(\Phi, s) = \sum_{i=1}^{|\theta|} q_{nb}(\phi_i) + s|\theta| - \sum_{i=1}^{|\theta|} q_0(\phi_i) - (1-s)|\theta| + \sum_{i=1}^{|\theta|} q_1(\phi_i)$. Because $q_{nb}(\phi_i) + q_1(\phi_i) = 1 - q_0(\phi_i)$, we can further simplify the equation to be $\mathcal{R}_{pass}(\Phi, s) = 2\sum_{i=1}^{|\theta|}\left(s - q_0(\phi_i)\right)$. Clearly, minimizing $\mathcal{R}_{pass}(\Phi, s)$ amounts to increasing $q_0(\phi_i)$ that leads to lower expected density and higher expected sparsity until $E_d = 1 - s, E_s < s$, which satisfies Lemma 1; (iii) **low expected density and high expected sparsity** (i.e., $E_d \leq 1 - s$ and $E_s \geq s$). Similarly, we can rewrite $\mathcal{R}_{pass}(\Phi, s) = 2\sum_{i=1}^{|\theta|}\left(1 - s - q_1(\phi_i)\right)$. Minimizing $\mathcal{R}_{pass}(\Phi, s)$ amounts to increasing $q_1(\phi_i)$ that brings down expected sparsity as well as increases expected density until $E_d < 1 - s, E_s = s$, which satisfies Lemma 1; It is impossible to have both **high expected density and expected sparsity** due to $E_s + E_d < 1$. We have shown that Lemma 1 holds in all possible cases. □

## B   Hyper-parameter Settings

We adopt an exponential escalation strategy to increase the value of regularization coefficient $\lambda$.

$$
\lambda = \lambda_{base} \cdot \lambda_0^{\#\text{train\_itr} / \#\text{n\_step}}
\tag{16}
$$

where $\lambda_{base}$ and $\lambda_0$ are hyper-parameters. We choose #n_step as $1,000$ in all experiments. We use the same inverse square root learning rate schedules for model training as in Li et al. (2021). Values of corresponding hyper-parameters are summarized in Table 6.

As for PASSCONC, $\lambda_c$ is defined as the minimal ratio between sparsifier gradients and concentrator gradients of $\phi_i \in \Phi$ for all heads to ensure that the concentrator is not dominated by the sparsifier while the

Table 6: Hyper-parameters

|  | ED Transformer | BERT-base |
|---|---|---|
| $\lambda_{base}$ | 1 | 1e-5 |
| $\lambda_0$ | 2 | 1000 |
| learning rate for $\Phi$ | 0.2 | 0.5 |

Table 7: Subnetwork performance at different sparsity levels, on MNLI and QQP.

| | Accuracy(MNLI) (%) | | | | | Accuracy(QQP) (%) | | | | |
|---|---|---|---|---|---|---|---|---|---|---|
| K | PASS | PASSCONC | DSP | LAG | Voita | PASS | PASSCONC | DSP | LAG | Voita |
| 16 | 79.38 | 78.38 | **80.69** | 78.19 | 80.14 | 89.18 | 88.50 | **89.25** | 86.87 | 85.53 |
| 32 | 80.73 | 79.63 | **82.64** | 81.33 | 80.30 | 89.94 | 89.36 | **90.09** | 88.99 | 89.99 |
| 48 | 82.59 | 82.51 | **82.92** | 82.80 | 82.03 | **90.57** | 90.40 | 90.16 | 89.67 | 90.20 |
| 64 | 83.23 | 83.06 | 83.55 | **83.74** | 75.70 | **90.80** | 90.74 | 90.28 | 90.02 | 87.21 |

concentration effect is modest to avoid damaging model training quality.

$$\lambda_c = \lambda \cdot \min \left\{ \left| \frac{\partial \mathcal{R}_{pass}}{\partial \phi_i} \right| \Big/ \left| \frac{\partial \mathcal{R}_{conc}}{\partial \phi_i} \right| \Big| \phi_i \in \Phi \right\} \tag{17}$$

$\lambda_c$ is adaptively updated in the middle of training, and set to 0 in both the early training phase and the last few training iterations to help improve model performance and achieve sparsity convergence. Specifically, $\lambda_c$ is set to 0 during the first $20,000$ iterations and the last $7,000$ iterations when training ED Transformer models. For BERT-base models, $\lambda_c$ is set to 0 except for iterations between $2,000$ and $5,000$.

As to the clipping range, there is a trade-off between choosing a larger range (e.g., [-10, 10]) and a smaller one (e.g., [-1, 1]). In general, a larger range allows us to better achieve subnetworks of desired almost-sure sparsities but the gradients will vanish to excessively small values (see Figure 3c) and make the gradient-descent optimization extremely slow. On the other hand, a small range allows efficient gradient-descent optimization but the corresponding gates may be neither almost surely closed nor opened (see Figure 3b) and therefore fail the almost-sure sparsity guarantees. For example, for a large range [-10, 10], the smallest gradient approaches 2e-5 (see Figure 3c) which leads to slow optimization rates, while for a small range [-1, 1], the maximal gate open/close probability is only 55% (see Figure 3b) that is neither almost surely opened nor closed. We use a held-out set to empirically select [-5, 5] as the clipping ranges because it gives a maximal gate open/close probability that is roughly 99% while the smallest gradient is above 0.003, which is practically sufficient for efficient optimization according to our experiments.

## C   More Performance Results

In this section, we provide detailed performance and speedup results on each of the 4 GLUE benchmark tasks: MNLI, QQP, QNLI, and SST-2 (see Section 4.1.1). Performance results are summarized in Tables 7 and 8, which resemble our analysis in Section 4.2. In general, PASS and PASSCONC are able to match or outperform all 3 baselines across a majority of experiment settings. Speedups achieved by different approaches are presented in Tables 9 and 10. Notably, PASSCONC outperforms all baselines in almost all cases which demonstrates its effectiveness in delivering efficiency improvements. Lastly, we report the Pareto front between model performance and speedups for subnetworks identified by different methods (see Figure 8), where PASSCONC dominates all other methods in most cases by achieving better performance-efficiency trade-offs, similar to our observation in Section 4.3.

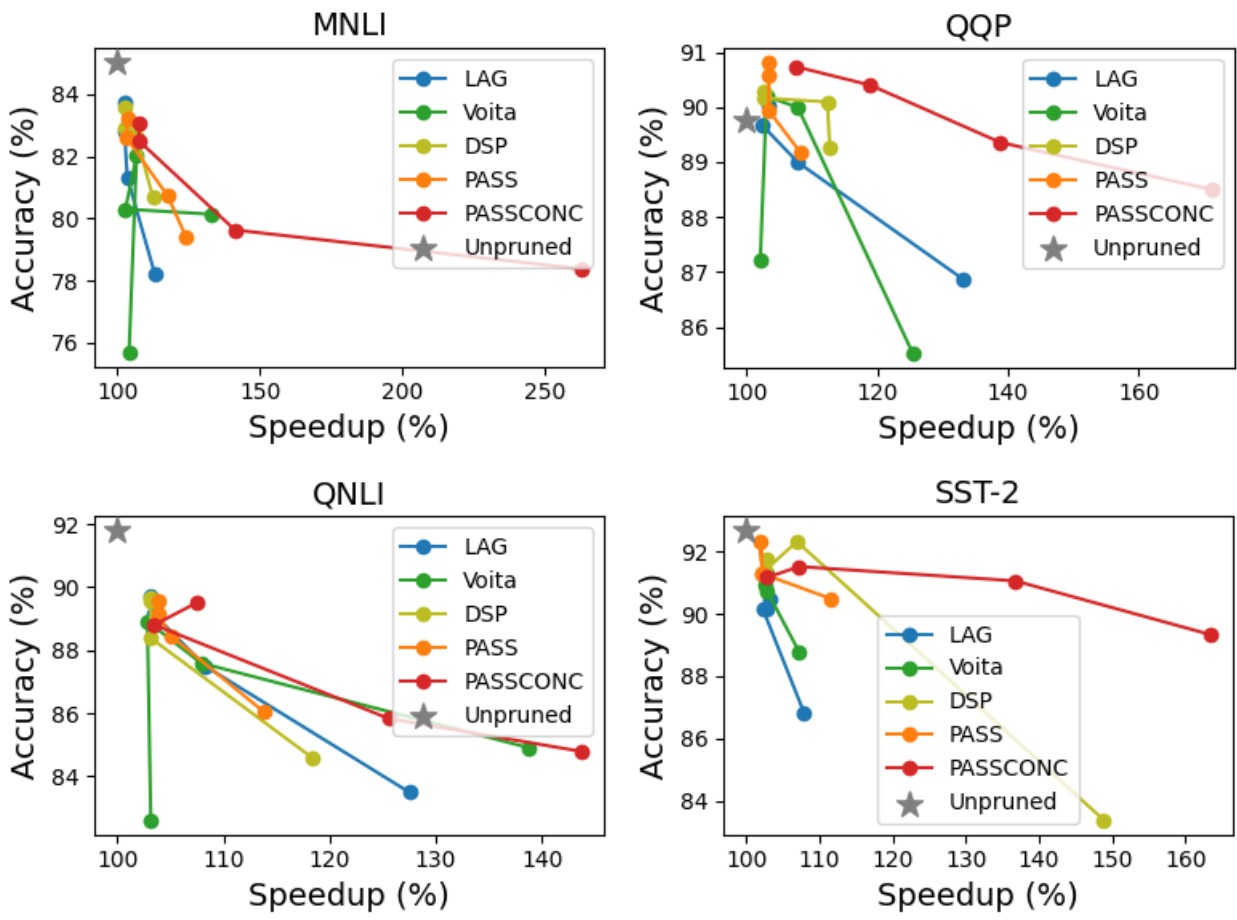

Figure 8: Attention speedups v.s. subnetwork performance on MNLI, QQP, QNLI, and SST-2. The unpruned model performance is included for reference purpose.

Table 8: Subnetwork performance at different sparsity levels, on QNLI and SST-2.

| | Accuracy(QNLI) (%) | | | | | Accuracy(SST-2) (%) | | | | |
|---|---|---|---|---|---|---|---|---|---|---|
| K | PASS | PASSCONC | DSP | LAG | Voita | PASS | PASSCONC | DSP | LAG | Voita |
| 16 | **86.03** | 84.79 | 84.59 | 83.49 | 84.90 | **90.48** | 89.33 | 83.37 | 86.81 | 88.76 |
| 32 | **88.43** | 85.83 | 88.38 | 87.50 | 87.59 | 91.28 | 91.06 | **92.32** | 90.14 | 90.71 |
| 48 | 89.13 | 88.80 | **89.58** | 89.13 | 88.91 | **92.32** | 91.51 | 91.40 | 90.48 | 90.94 |
| 64 | 89.57 | 89.51 | 89.68 | **89.69** | 82.61 | 91.28 | 91.17 | **91.74** | 90.14 | 91.28 |

Table 9: Attention speedups at different sparsity levels, on MNLI and QQP.

| | Speedup(MNLI) (%) | | | | | Speedup(QQP) (%) | | | | |
|---|---|---|---|---|---|---|---|---|---|---|
| K | PASS | PASSCONC | DSP | LAG | Voita | PASS | PASSCONC | DSP | LAG | Voita |
| 16 | 124.2 | **262.7** | 112.5 | 113.2 | 132.9 | 108.2 | **171.2** | 112.7 | 133.1 | 125.4 |
| 32 | 117.7 | **141.3** | 106.7 | 103.5 | 102.8 | 103.4 | **138.8** | 112.3 | 107.8 | 107.8 |
| 48 | 103.2 | **107.4** | 102.5 | 102.5 | 106.7 | 103.4 | **118.9** | 102.7 | 102.4 | 103.0 |
| 64 | 103.5 | **107.8** | 102.8 | 102.5 | 104.1 | 103.4 | **107.5** | 102.7 | 103.4 | 102.1 |

Table 10: Attention speedups at different sparsity levels, on QNLI and SST-2.

| | Speedup(QNLI) (%) | | | | | Speedup(SST-2) (%) | | | | |
|---|---|---|---|---|---|---|---|---|---|---|
| K | PASS | PASSCONC | DSP | LAG | Voita | PASS | PASSCONC | DSP | LAG | Voita |
| 16 | 113.8 | **143.7** | 118.3 | 127.5 | 138.8 | 111.5 | **163.3** | 148.8 | 107.9 | 107.2 |
| 32 | 105.1 | **125.6** | 103.1 | 108.2 | 107.9 | 102.2 | **136.7** | 106.9 | 102.2 | 102.8 |
| 48 | **103.8** | 103.5 | 103.1 | 103.5 | 102.8 | 101.9 | **107.2** | 102.5 | 103.1 | 102.5 |
| 64 | 103.8 | **107.5** | 103.1 | 103.1 | 103.1 | 102.2 | **102.8** | **102.8** | **102.8** | **102.8** |

