# OpenReview forum: "PASS: Pruning Attention Heads with Almost-sure Sparsity Targets"
_TMLR — Accepted by TMLR_

### Review · Reviewer_j5gS · 2024-07-06

**Summary Of Contributions:**

This paper introduces PASS and PASSCONC, two innovative methods for pruning attention heads in Transformer models. These methods leverage the concept of "almost-sure sparsity" to address the limitations of previous probabilistic pruning techniques, achieving desired sparsity levels with minimal performance degradation. Experiments on translation and GLUE benchmark tasks demonstrate that these methods outperform existing pruning techniques.

Contributions:
1. The authors propose a new concept termed "almost-sure sparsity," where the probability of a gate being open or closed approaches certainty. This ensures that the pruned model maintains high performance by avoiding the common threshold tuning problems associated with other pruning methods.

2. PASS: The paper introduces the Pruning with Almost-Sure Sparsity (PASS) framework, which enables effective pruning of attention heads during Transformer model training. This method involves learning a set of gate variables for each attention head, determining their likelihood of being pruned.

3.PASSCONC: To further enhance efficiency, the authors present a novel technique called "concentrator" within the PASS framework, named PASSCONC. This technique concentrates active attention heads in fewer layers, potentially reducing computational overhead significantly.

**Audience:**

Yes

**Claims And Evidence:**

Yes

**Requested Changes:**

To strengthen the work:

1. Include a more detailed discussion or guidelines on selecting hyperparameters, such as regularization coefficients and clipping ranges, for different tasks and models. This could involve providing heuristic methods, empirical results, or a systematic approach to tuning these parameters to achieve optimal performance.

2. The paper could benefit from a more thorough discussion on the applicability of these methods to other domains or more extensive models, for example decoder-only LLMs. This could include theoretical considerations or preliminary empirical results on diverse model architectures and tasks, highlighting the versatility and potential impact of the proposed methods.

3. A more detailed comparative analysis with other pruning methods, beyond just attention head pruning, could provide a clearer picture of the strengths and potential limitations of the proposed methods. This analysis could include comparisons with structured pruning, unstructured pruning, and other state-of-the-art techniques, supported by quantitative metrics and qualitative insights.

**Strengths And Weaknesses:**

Strengthes:
1. The paper introduces the concept of "almost-sure sparsity" and leverages it effectively for attention head pruning.

2. PASS and PASSCONC consistently outperform baseline methods across various sparsity levels and datasets, achieving both high accuracy and significant speedups.

3. The paper provides a clear mathematical formulation and justification for the proposed methods.

Weaknesses:

1. Although the evaluations are thorough for the presented tasks, the paper does not investigate the scalability of the methods to very large models or other language models, such decoder-based large language models (LLMs).

2. The success of the methods appears to depend on careful tuning of hyperparameters, such as the regularization coefficients and clipping ranges. The paper could benefit from providing more discussion or guidelines on selecting these hyperparameters for different tasks.

---

### Review · Reviewer_yuW5 · 2024-07-10

**Summary Of Contributions:**

This paper proposed a method for attention head pruning which incorporates user-specified sparsity target into objective function so that the desired sparsity can be achieved without additional hyper-parameter tuning. Furthermore, this paper introduced a second term into the objective function that encourages the unpruned heads to be localized in fewer layers. In another word, there will be a better chance that all heads in a certain layer, and this layer itself, could be pruned, and therefore, the computation of this entire layer could be skipped. To address potential training challenges, gradient clipping and gate reopening are employed. Finally, accuracy and speed-up are demonstrated on encoder-decoder Transformer and BERT, showing improvement over prior works.

**Audience:**

No

**Claims And Evidence:**

Yes

**Requested Changes:**

please see weakness above.

**Strengths And Weaknesses:**

Strengths:
1. sufficient experimental data to demonstrate the benefits over prior works.
2. Benefit of Concentrator is supported by Fig 5a. Benefit of clipping/reopening is supported by Fig 5b.

Weaknesses:
1. The reasoning of using an asymmetric design of probability function, i.e. q().
Compared to previous works, the choice of probability function is unclear. Intuitively, one would choose a probability function that enables "gate_close_probability = 1.0 - gate_open_probability" which naturally would result in a symmetric q(phi) about phi=0, such as used in Louizos 2018 Fig 2b. However, the author chose to use 2 separate probability functions for open and close and they are not complementary, as shown in Fig 2b of this manuscript. For example, when phi=0, q_open and q_close are both ~0.3, which further requires the introduction of an extra variable q_nb = 1 - q_open - q_close. Because of this, when trying to encourage q_0 to converge to sparsity s in Eq 6, q_1 will not automatically satisfy (1-s), hence the need of introducing an extra term in Eq 7 to resolve this. If the open and close probability were complementary for a given phi, as in Louizos, the objective function could be simplified greatly. Author may want elaborate on this seemingly unnatural choice.

2. Insight on the source of speed-up.
Depending on actual code implementation, head pruning could be translated into different computation schemes. One example (as in HuggingFace implementation) is that the pruned Linear layer will store a smaller weight tensor. And together with an additional indexing/slicing step on the incoming activation tensor, it can be used together with the pruned weights in the matmul. One important consideration here is that all heads in a layer, no matter how many there are, are computed at the same time. Splitting by heads and computing them one after another then concatenate would be very inefficient. In this scenario, speed-up would derive from performing matmul on smaller tensors, but the overhead of indexing activations is not negligible. It is unclear how authors implement their codes, but a brief discussion or profiling about the relative saving in compute vs additional overhead from indexing could give the readers a better idea how to benefit from head pruning schemes.

Additionally, according to Table 2 and Fig 5a, the main speed-up of this work comes from layer-pruning, not head-pruning, indicating the saving in computation is comparable with the overhead even when more than 50% of the heads were pruned (ie, BERT, keeping 64 out of 144 heads). But layer-pruning would be less preferrable as it will likely result in much more discernable accuracy degradation compared to fine-grained structured pruning at the same sparsity. (For example, https://developer.nvidia.com/blog/accelerating-inference-with-sparsity-using-ampere-and-tensorrt/). That being said, the reviewer would suggest to add some discussions, and experimental data if possible, to consider the memory-bound cases in addition to compute-bound cases. Especially for LLM, faster or reduced computation may not benefit as much because most of the time were spent on transferring weights and large tensors. If one considers pruning as a compression technique, under memory-bound situation, it might exhibit a much greater speed-up. A good example would be GPTQ in the quantization field.

---

### Review · Reviewer_wny5 · 2024-07-11

**Summary Of Contributions:**

The paper introduces a methodology (Pruning with Almost Sure Sparsity, PASS) for pruning attention heads in transformer models to achieve desired sparsity targets. This method relies on training head gates with an objective that ensures a binary open/close outcome for each gate. In addition, a concentrator is proposed which modifies the objective to encourage the unpruned heads to be concentrated in few layers as possible, so that entire layers can be pruned out (PASSCONC). A strategy for clipping and reopening gates is also introduced. Results on two transformer models show remarkable inference speedups with limited accuracy degradation, typically outperforming competing methods for attention head pruning.

**Audience:**

Yes

**Broader Impact Concerns:**

Not address in the manuscript. No concern on my end.

**Claims And Evidence:**

Yes

**Requested Changes:**

See Weaknesses in previous section.

**Strengths And Weaknesses:**

Strengths
- inference speed-up of transformers is a topic of clear interest to the audience of this journal
- the paper is clearly written, well structured, and the appendices offer meaningful support
- good theoretical treatment and justification of the proposed learning objective with almost-sure sparsity
- the custom learning objective for PASS, and the additional "concentrator" and "clipping and reopening" strategies appear to be novel
- the best combination of strategies results in remarkable speedups with limited accuracy degradation

Weaknesses
- clipping and gate reopening (quite effective in improving PASSCONC speed-ups) are always applied concurrently. What is the individual effect of these two distinct strategies? Have these been applied only to PASSCONC or to PASS as well?
- regarding clipping, $\phi_i$ is bound to a range of [-5,5], selected empirically. What is the impact of changing this threshold? Are the results sensitive to this arbitrary selection?
- in an effort to highlight which heads are most commonly pruned, Fig 6 averages the pruning frequencies across multiple sparsity settings and, for BERT, also across tasks; the averages are then sorted; however, without additional information on run-to-run variations, it's unclear what conclusions can be drawn from it; are the same heads consistently pruned as the sparsity target is raised? are they consistently pruned from one task to another?
- in addition, the text related to Fig 6b states that the heads in the middle layers (2-7) of BERT are more important and consequently pruned less frequently; this qualitative observation can't be easily derived from the figure itself; overall, I think a better visualization for Fig 6a,b or the use of a more concrete metrics for "overall head pruning" is warranted
- results are reported on just 2 models and limited tasks; it's unclear whether the proposed techniques are scalable to larger, more relevant models (admittedly, this limitation is mentioned by the authors as future work)

Other comments
- it should be specified that the model used is BERT-base; the manuscript does state it is a 12-layer BERT model and provides a citation to the original paper, but it's still worth using the common name for this model to facilitate the reader's understanding

---

### Decision · Action_Editor_BQjA · 2024-08-16

**Recommendation:** Accept with minor revision

**Comment:**

The paper proposes a method for attention head pruning which consists of training head gates variables with an objective that ensures the network has expected sparsity matching the desired target sparsity. The authors also propose another variant which introduces another term in the objective which encourages the unpruned heads to be concentrated in few layers, to enable pruning entire attention layers. They also employ a clipping and gate reopening strategy to address training challenges. Experiments on two types of transformer models (encoder-decoder and encoder-only) on a translation task and GLUE tasks show improvements over baselines.

Strengths:
- Paper is clearly written
- The proposed method PASS is motivated theoretically
- Experiments show that the proposed methods outperform baselines
- Ablation studies show the benefit of the additional concentrator loss and of the clip/reopening strategy on PASSCONC.

Weaknesses:
- No evaluation on large models or decoder-only models like LLMs.
- Proposed methods require tuning multiple hyper-parameters.
- Compute saving from pruning can be offset by indexing overhead on GPUs at medium to low sparsity.
- Missing implementation details for baselines and code not provided for now.
- It is unclear if the superior performance of PASS is due to the chosen sparsity objective or the clipping/reopening strategy or both. Need more ablation experiments to evaluate that.
- Important baseline missing.

The overall recommendations from the reviewers were mixed, with two reviewers leaning to accept and one leaning to reject. I summarized above the strengths and weaknesses highlighted by reviewers, as well as my own assessment. The authors revised the paper to address some of the reviewers comments but not all.  I recommend to "accept with minor revision". I request the following revisions to be made:
- Compare with the method of  [Voita et al., 2019] with clipping/reopening applied to it, to disentangle the benefit of clipping/reopening from the proposed sparsity objective.
- Compare with the CoFi method of [Xia et al, 2022], adapted to prune only heads and attention layers (so without pruning FFN layers). This method outperforms for example the method of [Wang et al, 2020c], which is the most recent baseline considered in the experiments. Alternatively, explain why you did not compare with this method.
- Release code to allow reproducing all empirical results.
- Include the discussion on the trade-off between saving in compute from pruning heads vs additional overhead from indexing and the table shown in the rebuttal (from your response to Reviewer yuW5). Clarify the apparent mismatch between the results in the paper showing speedups on a GPU with up to K=64 unpruned heads, and the ones in the new table showing that speedups on a GPU are only achieved up to k=4 unpruned heads!
- Include the discussion on how you chose the clipping range and on the importance on applying clipping and gate reopening jointly (from your response to Reviewer wny5).
- Provide more details on how you implemented the baselines.

**Audience:**

yes but limited. Results on LLMs and larger models would greatly increase the interest in this paper.

**Claims And Evidence:**

Yes for some claims but not all. More evidence is needed to back the claim of outperforming SOTA on the considered models and tasks. See comments below.